# Neuroprotective Effects of Krypton Inhalation on Photothrombotic Ischemic Stroke

**DOI:** 10.3390/biomedicines12030635

**Published:** 2024-03-13

**Authors:** Viktoriya V. Antonova, Denis N. Silachev, Egor Y. Plotnikov, Irina B. Pevzner, Elmira I. Yakupova, Mikhail V. Pisarev, Ekaterina A. Boeva, Zoya I. Tsokolaeva, Maxim A. Lyubomudrov, Igor V. Shumov, Andrey V. Grechko, Oleg A. Grebenchikov

**Affiliations:** 1Federal Research and Clinical Center of Intensive Care Medicine and Rehabilitology, Moscow 107031, Russia; pisarev@gmail.com (M.V.P.); ball.87@mail.ru (E.A.B.); tsokolaevazoya@mail.ru (Z.I.T.); mlyubomudrov@fnkcrr.ru (M.A.L.); shumoff-03@mail.ru (I.V.S.); oleg.grebenchikov@yandex.ru (O.A.G.); 2A.N. Belozersky Institute of Physico-Chemical Biology, Lomonosov Moscow State University, Moscow 119992, Russia; silachevdn@genebee.msu.ru (D.N.S.); plotnikov@belozersky.msu.ru (E.Y.P.); irinapevzner@mail.ru (I.B.P.); elmira.yaku@gmail.com (E.I.Y.)

**Keywords:** krypton, noble gas, neuroprotection, stroke, photothrombotic ischemic stroke, Nrf2, GSK, AKT, NFkB

## Abstract

This is the first in vivo study to investigate the neuroprotective effects of krypton on focal cerebral ischemia. The aim of the study was to analyze the effect of 2 h of inhalation of a krypton–oxygen mixture (Kr 70%/O_2_ 30%) on the recovery of neurological functions and the degree of brain damage in rats after photoinduced ischemic stroke (PIS) and to investigate the possible mechanisms responsible for this neuroprotection. Experiments were performed on male Wistar rats weighing 250–300 g (n = 32). Animals were randomized into four groups. Two groups (n = 20) underwent photoinduced ischemic stroke, followed by 2 h of inhalation of krypton–oxygen mixture consisting of Kr 70%/O_2_ 30% or a nitrogen–oxygen breathing mixture consisting of N_2_ 70%/O_2_ 30%, followed by neurological examinations on days 3 and 7. The other two groups (n = 12) received only gas mixtures of the same concentration and exposure duration as in those in the PIS groups, then Western blot analysis of the potential molecular mechanisms was performed. The results of the study show that treatment with the krypton–oxygen mixture consisting of Kr 70%/O_2_ 30% improves the neurological status on day 7 of observation, reduces the lesion volume according to the MRI examination and the number of Iba-1- and caspase-3-positive cells in the damaged area, promotes the activation of neoangiogenesis (an increase in the von Willebrand factor), and reduces the penumbra area and the number of NeuN-positive cells in it on day 14 of observation. Inhalation of the krypton–oxygen mixture also significantly increases the levels of phosphorylated AKT kinase (protein kinase B) and glycogen synthase kinase 3b (pGSK3b) and promotes the expression of transcription factor Nrf2, which was accompanied by the lowered expression of transcription factor NFkB (p50). Thus, we showed pronounced neuroprotection induced by krypton inhalation after stroke and identified the signaling pathways that may be responsible for restoring neurological functions and reducing damage.

## 1. Introduction

For several decades, there has been a growing interest in the use of noble gases in the practice of medicine. Noble gases can exert biological effects on living organisms without being metabolized since they are unable to undergo chemical reactions with other elements of the Mendeleev Periodic Table or with biological molecules. In clinical medicine, the anesthetic and neuroprotective effects of noble gases are of high interest, and their mechanisms are being investigated in both experimental and clinical studies [1].

The most commonly used inert gas in clinical practice is xenon; however, researchers have recently turned their attention to other inert gases that do not have anesthetic properties but may have similar neuroprotective properties. These include argon and krypton.

Several studies [2,3,4,5] have shown that argon has neuroprotective properties in ischemic models, which are presumably related to the signaling pathway of the transcription factor Nrf2 (factor-2, also known as the erythroid nuclear factor). Nrf2 regulates the response to oxidative stress, the redox balance, antioxidant enzymes, and the resistance of cells to various stressors [5,6]. However, several studies have provided conflicting data on the neuroprotective properties of argon in traumatic brain injury (TBI) [7,8,9], leaving open the question of its potential use as a universal neuroprotectant.

The second inert gas that has been proposed to protect against acute brain injury is krypton. Although its biological properties have not been extensively studied [10], krypton has been used in several experimental studies in hypoxic gas mixtures to increase survival during induced normobaric hypoxia [10,11]. Compared with controls and rats preconditioned with an argon–oxygen mixture, rats preconditioned with a normoxic gas mixture containing 79% krypton were better able to tolerate exogenous normobaric hypoxia with a lower oxygen content of up to 4%. The presence of krypton in the hypoxic gas mixture reduced oxygen consumption to normoxic levels [12], which was confirmed by both instrumental (vital sign monitoring) and laboratory studies (the increase in deoxyhemoglobin was less significant in the krypton group, suggesting that tissue oxygenation was maintained close to normoxic levels).

While krypton is not an anesthetic at a normal pressure [11], it has anesthetic properties when administered at an elevated pressure [11] that can be observed for both animals and humans. This can be useful in medical scenarios where conventional anesthetics are not suitable due to allergic reactions or other limitations. In addition, krypton has been found to act as a sedative by enhancing GABA [13], further expanding the range of potential medical applications. Gamma-aminobutyric acid (GABA) is the most important inhibitory neurotransmitter in the brain. By enhancing GABA-ergic activity, krypton can provide a sedative effect without relying on conventional sedative drugs, which can be addictive or have other side effects.

In this study, the neuroprotective properties of krypton were evaluated using an experimental model of acute ischemic brain injury. We specifically investigated the effects of a 2 h inhalation of a krypton–oxygen mixture consisting of 70% krypton (Kr) and 30% oxygen on both the recovery of neurological function and the reduction of cerebral damage in rats exposed to photochemically induced ischemic stroke. The possible mechanisms underlying the observed neuroprotective effects were also investigated.

## 2. Materials and Methods

### 2.1. Experimental Animals

The experiments were performed on male Wistar rats weighing 250–300 g (n = 32). Animals were deprived of food for 6 h before the experiment, but they had free access to water. The study protocol was approved by the Local Ethical Committee of the Center of Intensive Care Medicine and Rehabilitology (Moscow, Russia) No. 3/22/3, dated 14 December 2022. Experiments were performed in accordance with the requirements of Directive 2010/63/EU of the European Parliament and Council of the European Union on the protection of animals used for scientific purposes.

Photothrombotic ischemic stroke was modeled in 20 animals. The animals were randomly divided into two groups as follows: a control group with photothrombotic ischemic stroke (PIS) followed by exposure to a nitrogen–oxygen gas mixture consisting of 70% N_2_/30% O_2_ (Stroke + N_2_ group) (n = 10) and an experimental group with PIS followed by exposure to a krypton–oxygen gas mixture consisting of 70% Kr/30% O_2_ (Stroke + Kr group) (n = 10).

To study the signaling pathways responsible for neuroprotection, a series of experiments was performed on 12 animals randomly divided into two groups according to the atmosphere in which they were housed as follows: a control group inhaling a nitrogen–oxygen mixture consisting of 70% N_2_/30% O_2_ (N_2_ group, n = 6) and an experimental group inhaling a krypton–oxygen mixture consisting of 70% Kr/30% O_2_ (Kr group, n = 6). No additional interventions were performed in this series. Photothrombotic ischemic stroke modeling was not performed.

### 2.2. Modeling of Photothrombotic Ischemic Stroke

After an intraperitoneal injection of 6% chloral hydrate at a dose of 300 mg/kg [14], photothrombotic ischemic stroke was modeled according to a previously described protocol [15]. Focal ischemic stroke was modeled in the sensorimotor cortex of the rat brain using photochemical cortical vascular thrombosis. The photosensitive dye Rose Bengal (3%, 40 mg/kg intravenously; Sigma-Aldrich, St. Louis, MI, USA) was injected into the jugular vein. The skull was exposed through a midline incision, and the periosteum was removed. The sensorimotor cortex region (stereotactic coordinates relative to the bregma are 0.5 mm distally and 2.5 mm laterally) was then irradiated with green light at λ = 550 nm (3 mm in diameter) for 15 min. After skin suture, the rats were placed in a cage under an infrared heating lamp until they recovered from anesthesia. Body temperature was maintained at 37 ± 0.5 °C throughout the experiment using a heating pad with a feedback sensor.

Mortality in the animal groups was assessed at 24 h, 7 days, and 14 days after stroke. No deaths occurred in the experimental animals. No humane endpoint was reached during the 14-day observation period, and no animals were withdrawn from the study.

### 2.3. Exposure to Krypton

Ninety minutes after PIS, the animals were placed in a 35 L plastic chamber in which a fresh gas mixture (70% N_2_/30% O_2_ for the Stroke + N_2_ group; 70% Kr/30% O_2_ for the Stroke + Kr group) was continuously supplied at a flow rate of 0.5 L/min per animal. No more than 5 animals were in the chamber at any one time to avoid hypoxia and hypercapnia. The exposure time in the chamber was 2 h. Throughout the experiment, the O_2_ and CO_2_ levels in the chamber with the animals were continuously monitored with a gas analyzer (INSOVT, Saint-Petersburg, Russia). At the end of the exposure period, the general condition of the animal (level of alertness and mobility) was assessed, and anesthesia (paracetamol 50 mg/kg subcutaneously) was administered. The animal was then placed in a cage with free access to water and food. Animals in groups N_2_ and Kr underwent a similar procedure of inhalation of gas mixtures but without anesthesia and modeling of PIS.

### 2.4. Assessment of Neurological Status

The neurological status of the animals was assessed on days 3 (D3) and 7 (D7) after ischemia induction using the limb-placing test (LPT). The protocol used was based on the method described by De Rik et al. [16] and modified by Jolkkonen et al. [17]. Rats were hand-trained for one week prior to testing. The test consisted of seven tasks assessing sensorimotor integration of the forelimbs and hindlimbs in response to tactile, proprioceptive, and visual stimuli. Each task was scored as follows: complete performance without delay, 2 points; delayed (>2 s) and/or incomplete performance, 1 point; and failed performance, 0 points. The scores were summed, and the results were presented as the sum of the task scores.

### 2.5. Brain MRI

On day 14 after stroke, the animals underwent MRI in a 7 Tesla magnetic field induction tomograph with a gradient system of 105 mT/m (BioSpec 70/30, Bruker, Germany). The rat was anesthetized with isoflurane (1.5–2 vol%) and placed in a stereotaxic and thermoregulatory device, as previously described [18].

A standard protocol was used to study the rat brain, including the acquisition of T2-weighted images. The imaging protocol included a T2-weighted image sequence (time to repetition = 4500 ms; time to echo = 12 ms; slice thickness = 0.8 mm). A standard protocol was used to study the rat brain, including the acquisition of T2-weighted images. The total scanning time per animal was approximately 25 min. The extent of brain damage was assessed by graphical analysis of the MR images by calculating the volume of the brain lesion. For this purpose, the lesion area in mm^2^ per slice was calculated on a series of MR images using ImageJ software (National Institutes of Health Image software, Version 1.8.0, Bethesda, MD, USA). For this purpose, the areas of unaltered tissue in the healthy (S1) and damaged (S2) hemispheres were identified separately, and the lesion area was calculated using the formula ΣS = S1 − S2, where ΣS is the lesion area per slice (mm^2^). The brain lesion volume was calculated using the formula V = ΣSn × d, where d is the thickness of a slice (0.8 mm) and ΣSn is the sum of the lesion areas on all slices (mm^2^) [18].

### 2.6. Immunohistochemical Study

For immunohistochemistry, rat brains were fixed in 10% formalin, embedded in paraffin, and sectioned at 4 μm. Deparaffinization and antigen unmasking of the paraffin sections were performed with commercial Trilogy^®^ Pretreatment Solution Cell Marque according to the manufacturer’s protocol. Sections were incubated in 2% hydrogen peroxide for 10 min to inhibit endogenous peroxidase. Protein Block (ab64226, Abcam, Cambridge, UK) was used for 30 min in a humidified chamber to prevent the nonspecific binding of primary or secondary antibodies to tissue proteins. Sections were incubated for 1 h in a humidified chamber at 37 °C with primary antibodies including the NeuN antibody (ab177487 1:200, Invitrogen, CA, USA), anti-caspase-3 (ab13847 1:100, Invitrogen), the anti-von Willebrand factor (ab 9378 1:200, Abcam), and anti-Iba1 (ab5076, 1:500, Abcam) diluted in an antibody diluent (ab64211, Abcam). Secondary antibodies Dako REAL EnVision Detection System (DAB Dako Antibody Diluent) or ImmPACT^®^ Vector^®^ Red Substrate Kit, Alkaline Phosphatase (AP) (SK-5105) were used according to the manufacturer’s protocol. Sections were stained with hematoxylin for 1–2 min and then dehydrated in 70%, 96%, 100% alcohol and cleared twice in xylene. Images were captured on an Aperio CS2 microscope (Leica, CA, USA), and digital analysis was performed using NIS-Elements (NIS-Elements Basic research Nikon Corporation, Version 5.20.00, Tokyo, Japan) and ImageJ (National Institutes of Health Image Software, Version 1.8.0, Bethesda, MD, USA) software.

### 2.7. Western Blot Analysis

The animal’s brain hemispheres were excised 24 h after inhalation and homogenized in PBS (pH 7.4) (Amresco, Cleveland, OH, USA) with 1 mM protease inhibitor PMSF (Amresco). Protein concentration was measured by bicinchoninic acid assay (Sigma, St. Louis, MO, USA). Samples of brain homogenates were loaded onto the gradient (5–20%) Tris-glycine polyacrylamide gels (10 µg total protein per lane). After electrophoresis, the gels were blotted onto PVDF membranes (Amersham Pharmacia Biotech, Newcastle, UK). Membranes were blocked with 5% (wt/vol) nonfat milk (SERVA, Heidelberg, Germany) in PBS containing 0.05% (vol/vol) Tween 20 (Panreac, Barcelona, Spain) and then incubated with primary antibodies including rabbit polyclonal anti-Akt 1:1000 (#9272, Cell Signaling, Danvers, MA, USA), rabbit polyclonal anti-phospho-Akt 1:700 (#4060, Cell Signaling, Danvers, MA, USA), rabbit polyclonal anti-phospho-GSK3b 1: 500 (#9336, Cell Signaling, Danvers, MA, USA), mouse polyclonal anti-GSK3A/B 1:1000 (#sc-9271, Santa Cruz Biotech, Dallas, TA, USA), rabbit polyclonal anti-Nrf2 1:1000 (#ab137550, Abcam, Cambridge, UK), rabbit polyclonal anti-NFkB (p50) 1:1000 (#sc-114, Santa Cruz Biotech, USA), mouse monoclonal anti-b-actin 1:2000 (#A2228, Sigma, USA), and mouse monoclonal anti-GAPDH 1:2000 (#5G4cc, Hytest, Moscow, Russia).

Membranes were stained with secondary anti-rabbit IgG or anti-mouse IgG conjugated with horseradish peroxidase 1:5000 (both IMTEK, Moscow, Russia). Specific bands were visualized using an Advansta Western Bright™ ECL kit (Advansta, San Jose, CA, USA). Detection was performed using a V3 Western Blot Imager system (BioRad, Hercules, CA, USA) and band density was measured using Image Lab software (BioRad, Version 5.1, Hercules, CA, USA). B-actin was used as an internal control, and total AKT and GSK3a/b proteins were used for the signal normalization of the phosphorylated form of the proteins.

### 2.8. Statistical Analysis

The accumulation and the primary analysis of the data were carried out using Microsoft Office Excel 2019. Statistical analysis was performed using SPSS Statistics software (IBM SPSS Statistics for Windows, Version 27.0.1, Ar-monk, IBM Corp, New York, NY, USA) and GraphPad Prism software (GraphPad Software, Version 8.0.1, Boston, MA, USA) The normality of the parameter distribution was assessed using the Shapiro–Wilk criterion. Descriptive statistics of the quantitative data are presented in Me (Q1; Q3) format, where Me is the median, Q1 is the first quartile (25th percentile), and Q3 is the third quartile (75th percentile), or the mean and standard deviation. The non-parametric Mann–Whitney U test was used for the comparative intergroup analysis of the quantitative independent variables. The statistical significance level was set at *p* < 0.05.

## 3. Results

### 3.1. Krypton Protects Rat Brains Exposed to PIS

Evaluation of neurological status based on the LPT showed that rats after PIS had severe neurological deficits during all periods of the experiment (Figure 1). The neurological status of intact rats was 14 points before brain damage modeling, whereas after PIS modeling it was approximately 7 points on day 3 and remained so until day 7 of observation.

On day 3 (point D3) after PIS modeling, there was no significant difference in neurological status between the Stroke + Kr and Stroke + N_2_ groups (*p* = 0.4020). On day 7 after stroke, there was a significant improvement in neurological status in the Stroke + Kr group (*p* = 0.0245), which may indicate the faster neurological rehabilitation of the animals in this group (Figure 1a).

MRI revealed a clearly visible brain lesion appearing as an area of the hyperintense signal in the sensomotor cortex down to the striatum (Figure 1c).

Morphometric analysis of the MR images showed that the brain lesion volume was significantly lower in the Stroke + Kr group than that in the Stroke + N_2_ group, averaging 17.56 [14.18; 20.38] mm^3^ and 22.19 [17.98; 24.42] mm^3^, respectively (*p* = 0.0435) (Figure 1b).

### 3.2. Krypton Inhibits Neuroinflammation and Cell Death after Ischemic Stroke

In both groups, immunohistochemical staining revealed NeuN-positive cells throughout the ischemic lesions. In addition, these cells were detected in the center of the lesion. Beyond the Penumbra border, neurons showed intense staining with antibodies against NeuN (Figure 2c,d). There were no significant differences in the number of NeuN-positive cells between the nitrogen and krypton groups (*p* = 0.8254). The mean number of neurons was 228.2 ± 178.7/mm^2^ and 27.2 ± 11.88/mm^2^ in the Stroke + Kr and Stroke + N_2_ groups, respectively (Figure 3a).

Immunohistochemical detection of the microglial marker Iba-1 (ionized calcium-binding adaptor molecule 1) revealed a larger area to be occupied by Iba-positive cells in the Stroke + N_2_ group compared with that in the Stroke + Kr group (*p* = 0.0476) (Figure 3c). Meanwhile, these cells, which we refer to as macrophages, were mainly localized in the necrotic zone (Figure 2a,b).

The mean penumbra width was significantly lower in the Stroke + Kr group (0.06 [0.05; 0.073] mm) than in the Stroke + N_2_ group (0.09 [0.07; 0.09] mm) (Figure 3e). The number of NeuN-positive cells in the penumbra was higher in the Stroke + Kr group (*p* = 0.0481) (Figure 3b). The ratio of Iba-positive cells to the lesion area showed no significant difference (*p* = 0.0952) (Figure 3d). This suggests that reduced macrophage numbers are primarily due to a lower lesion volume rather than reduced lesion infiltration.

The ratio of the area of the von Willebrand factor (vWF)-positive staining to the total lesion area in the Stroke + Kr group was almost twice that in the control group, with 0.11 (0.10; 0.13) versus 0.06 (0.04; 0.07) (*p* = 0.029) (Figure 3g). Furthermore, in the Stroke + Kr group, the vWF was equally detected in the penumbra and lesion, which may indicate active neoangiogenesis in the lesion zone compared with that in the control group, in which the vWF was predominantly detected in the penumbra (Figure 2e,f). A significant decrease in Cas-3-positive cells (*p* = 0.032) was observed in the Stroke + Kr group compared with that in the Stroke + N_2_ group, with a distribution density (the ratio of the number of Cas-3-positive cells to the total lesion area) of 3672 (3158; 4448), with a.u./mm^2^ vs. 5712 (4512; 8574) and a.u./mm^2^ in the experimental and control groups, respectively (Figure 3f).

### 3.3. Molecular Mechanisms of Neuroprotection

Analysis of the key signaling pathways associated with ischemic tolerance showed that the amount of the phosphorylated form of AKT kinase (protein kinase B) in brain tissue increased significantly (*p* = 0.0012) when the animals were exposed to krypton for 2 h. However, the total amount of the AKT protein did not change. Similarly, a trend toward increased levels of phosphorylated GSK3b kinase was observed in the brain, which are downstream in the signaling cascade and associated with preconditioning [19,20] The signal for pGSK3b was normalized to the total GSK3b protein, whose levels were also unchanged after exposure to krypton.

In addition, an increase in the expression of the transcription factor Nrf2, which regulates the cellular response to oxidative stress, was observed in the brains of rats exposed to krypton. The transcription factor NFkB (p50) was also downregulated after krypton exposure, thereby indicating anti-inflammatory changes in brain tissue induced by krypton (Figure 4).

## 4. Discussion

In our study, we demonstrated for the first time that the noble gas krypton has neuroprotective effects against focal ischemic brain injury. Similar studies have been performed on other noble gases; however, it is not yet possible to conclude that their neuroprotective effects are universal. The neuroprotective properties of noble gases have been studied for a long time, but the data obtained to date are contradictory. For example, many in vitro and in vivo studies convincingly demonstrate the protective effects of xenon on various brain diseases [21,22]. In addition, the mechanisms underlying xenon-induced neuroprotection have been extensively studied [23,24]. For argon, there are far fewer studies [22,25,26], and they do not always clearly indicate the presence of protective effects on the brain [8,9,21,27]. Few studies have demonstrated the neuroprotective potential of neon and krypton [24]. Moreover, like argon, they often do not show a significant protective effect, e.g., in cell cultures [28,29]. Obviously, neuronal culture models cannot reproduce all the pathological mechanisms that develop in the ischemic brain, and they do not encompass all the potential protective pathways. Therefore, in our study, we used a model of brain ischemia that most closely resembles the clinical situation of focal ischemic stroke.

In a study conducted on a photochemical ischemic stroke model based on focal endothelial injury with subsequent thrombosis of rat cortical arteries, krypton caused the recovery of sensorimotor functions on day 7 and a reduction in the volume of damage on day 14.

In the Stroke + Kr group, we observed a reduction in the lesion area and penumbra width as well as an increase in the number of NeuN-positive cells in the penumbra zone, thereby indicating the number of neurons. In addition, we observed a decrease in the level of Iba-1, which is a calcium-binding protein specific to microglia and macrophages and is involved in phagocytosis and is considered a traditional marker of activated microglia. A decrease in these markers could indicate less active inflammation in the injury zone, which correlated with a reduction in the volume of the cortical injury and consequently a reduction in the severity of the neurological deficit.

Another important finding was the difference in the level of the vWF, which plays an important role in both hemostasis and angiogenesis according to recent data [30,31,32]. Its significant increase in the penumbra zone and lesion focus in the Stroke + Kr group compared with the Stroke + N_2_ group together with other signs of the faster repair of necrotic lesions may indicate the activation of neoangiogenesis in the Stroke + Kr group. This process underlies the earlier restoration of blood flow in the stroke area, which contributes to early rehabilitation after a cerebrovascular accident [33]. The decrease in the number of Cas3-positive cells only confirms the role of apoptosis inhibition, namely that AKT phosphorylation is a possible molecular mechanism of neuroprotection by krypton [34,35].

Studies on the neuroprotective effects of xenon and argon have identified several key mechanisms that provide brain protection [24,36,37,38] We believe that the most important mechanisms are those of preconditioning-related signaling and the regulation of the redox state. In this context, while studying the putative molecular pathways involved in the neuroprotective effect of krypton, we tested the hypothesis of its effect on the signaling cascades involved in the universal mechanisms of organ protection. Indeed, an increase in the phosphorylated form of GSK-3β and AKT involved in the preconditioning cascade has been demonstrated [37]. Numerous studies have shown that GSK-3β plays a key role in the mitochondrial permeability transition, which under ischemic conditions triggers pathological cascades leading to mitochondrial dysfunction, oxidative stress, and cell death. Inhibition of this enzyme by Ser-9 phosphorylation results in the inhibition of mitochondrial pore opening and prevents the downstream signaling cascade leading to apoptosis. AKT kinase is upstream of GSK-3β in this signaling cascade, and its activation by phosphorylation also strongly suggests the activation of signaling to increase brain tolerance to ischemia. Similar changes in this kinase cascade were observed for argon and xenon treatment, thus suggesting a common mechanism of action between these noble gases and krypton.

The second identified mechanism of krypton-induced neuroprotection was that of an increase in the level of the Nrf2 protein, which plays an important role in coordinating the transcription of proteins responsible for cellular defense against oxidative stress. This transcription factor is a master regulator of the cellular response to oxidative stress, thereby triggering the expression of many antioxidant enzymes and proteins involved in maintaining cellular redox homeostasis [39]. Its activation usually occurs in response to an increase in oxidants and has been observed during both ischemia and ischemic preconditioning through the degradation of Keap-1 and the stabilization of Nrf2 [40]. Another pathway of Nrf2 regulation is that of its phosphorylation by GSK-3 kinase, which causes Nrf2 activation and the antioxidant response of the cell when inhibited, as seen after exposure of the rat brain to krypton.

Finally, the third mechanism of action of krypton on the ischemic brain could be that of the reduction in the inflammatory response that inevitably occurs in stroke, both in the ischemic focus and in the adjacent tissues. Indeed, the amount of the NFkB factor, which serves as an important marker for a proinflammatory shift in the tissue, decreased in the brain tissue. This mechanism was also confirmed by the cellular responses to the photothrombosis stroke model—a decrease in macrophage infiltration evidenced by a decrease in Iba-positive cells. It should be noted that the regulation of NF-kB is in some cases associated with the effects of Nrf2 [41,42,43], again confirming the influence of krypton action on the totality of these protective mechanisms.

This study has some limitations that should be considered. First, the chosen exposure duration was determined using previous research on xenon and assuming that these inert gases have similar biological effects. However, the underlying mechanisms of action of these gases may be different, so the determination of an optimal exposure time is an unresolved issue. Second, our investigation was limited to exploring selected potential mechanisms of neuroprotection in the intact rat brain. Given the absence of comparable studies, our results may provide valuable insights for other researchers in this field. However, it is important to recognize that the verification of these initial findings in models of acute brain injury is a critical next step.

The current lack of data on the efficacy and safety of krypton as a neuroprotective agent in focal cerebral ischemia does not allow definitive conclusions to be drawn about its prospects for future use. Our study is the first to suggest a high potential for the use of this gas in acute cerebrovascular events. Based on our results and the available research data on the neuroprotective effects of other noble gases such as xenon, argon, and helium [1,44], we emphasize the importance and necessity of further research on the clinical use of krypton-based gas mixtures.

## Figures and Tables

**Figure 1 biomedicines-12-00635-f001:**
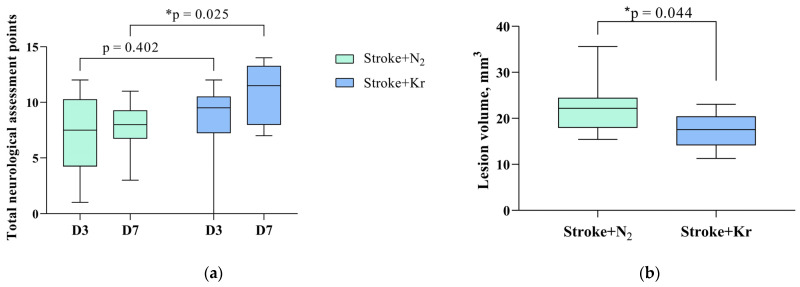
Krypton treatment improves neurological status and reduces brain damage after stroke. (**a**) Effect of krypton on neurological status determined by a limb-placing test up to 7 days after PIS; (**b**) Lesion volume assessed by MRI on day 14 after PIS; (**c**) Representative T2-weighted MR images of series coronal brain slices taken 14 days after PIS in the comparison groups. Data are presented as median with interquartile range (Q1; Q3). * Mann–Whitney U test, *p* < 0.05.

**Figure 2 biomedicines-12-00635-f002:**
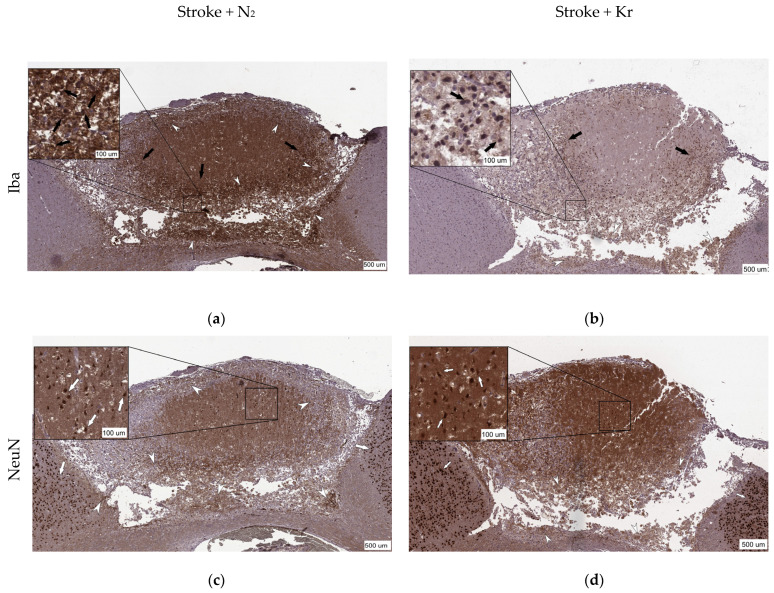
Representative images of the immunohistochemical staining of the brain sections after photothrombosis (**a**,**c**,**e**) and after photothrombosis with krypton treatment (**b**,**d**,**f**). (**a**,**b**) Positive staining for Iba-1, a marker of activated immune cells (black arrow) in the damaged area, and the accumulation of foam cells (white arrowhead); (**c**,**d**) staining for NeuN-positive cells (white arrow) and the accumulation of foam cells (white arrowhead). NeuN-positive staining indicates the number of neurons in the lesion area and in the penumbra but not the viability of these neurons; (**e**,**f**) double staining for the marker of caspase-3 (brown) and the von Willebrand factor (red); Cas-3-positive staining indicates apoptotic cells (black asterisk) and vWF-positive staining indicates neoangiogenesis (black arrowhead).

**Figure 3 biomedicines-12-00635-f003:**
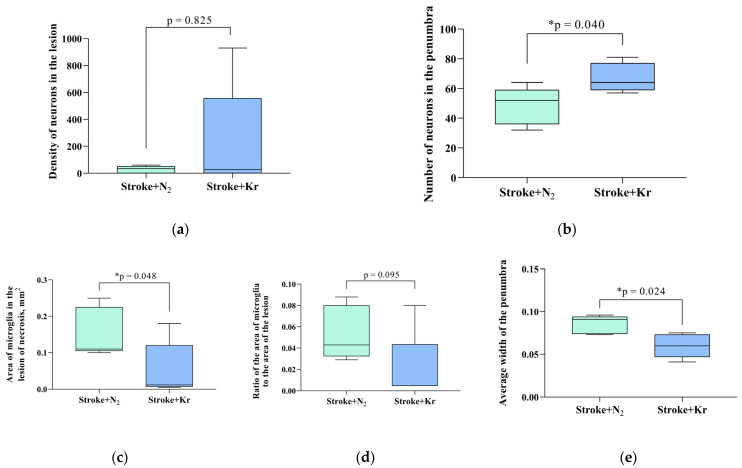
Pathological changes in the rat brain after stroke and the effects of krypton inhalation. (**a**) Neuron density in the ischemic area and (**b**) number of neurons in the penumbra area; (**c**) microglial area in the lesion; (**d**) the ratio between the microglia-positive area and the area of the lesion; (**e**) mean penumbra width; (**f**) density of Cas3-positive cells in the lesion area; (**g**) ratio of the vWF area to the total lesion area. Data are expressed as the median and interquartile range (Q1; Q3). * Mann–Whitney U test, *p* < 0.05.

**Figure 4 biomedicines-12-00635-f004:**
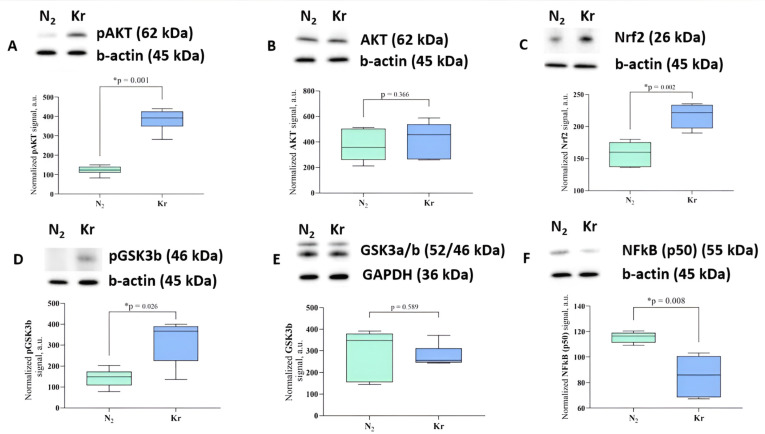
Changes in signaling pathways in the rat cerebral cortex after krypton exposure. Representative immunoblots and densitometry for (**A**) pAKT. (**B**) AKT, (**C**) Nrf2, (**D**) pGSK3b, (**E**) GSK3b, and (**F**) NFkB (p50) with b-actin or GAPDH as the loading control, n = 6. The observed molecular weights of the proteins are labeled. Data are expressed as the median and interquartile range (Q1; Q3). * Mann–Whitney U test, *p* < 0.05. Raw uncropped western blot images are provided in Appendix A.

## Data Availability

Data can be provided on request.

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
