# Peer review of "Neuroprotective Effects of Krypton Inhalation on Photothrombotic Ischemic Stroke"

_biomedicines, 2024, doi:10.3390/biomedicines12030635_

Round 1

Reviewer 1 Report

Comments and Suggestions for Authors

Thank you sincerely for allowing me to review the article titled "Neuroprotective effects of krypton inhalation in photothrombotic ischemic stroke" (biomedicines-2869486), submitted to the Special Issue: Brain Injury: New Insights into Mechanisms and Future Promising Treatments.

The study's primary objective was to assess the impact of a 2-hour inhalation of a krypton-oxygen mixture (Kr 70%/O2 30%) on the recovery of neurological functions and the extent of brain damage in rats following photoinduced ischemic stroke (PIS). Additionally, the investigation aimed to explore potential mechanisms contributing to the observed neuroprotection.

Within this study, the neuroprotective properties of krypton were scrutinized using an experimental model of acute ischemic brain injury.

In the results section, on page 7, graphical representations lacking captions and enumeration are presented. To enhance clarity and facilitate referencing in the text, it is recommended to provide labels and numbering for these graphs. Moreover, I propose breaking down Figure 1 into its distinct components for improved comprehension, extending the same suggestion to Figures 3 and 4.

The discussion is notably insightful. However, it may benefit from a thorough consideration of potential limitations in the experimental work conducted and the necessity for confirmation in human subjects. It would also be valuable to explore future research directions and the potential applicability of the findings in future healthcare settings.

Author Response

Dear Editors, 

Thank you for giving us a chance to revise our manuscript. We have modified the text to answer all recommendations expressed in the reviewer’s comments. We have made the appropriate changes in the text and the figures. We appreciate the constructive criticism and believe that after these revisions, our manuscript will not raise more critique.

Thank you for your work.

We have attached the answers to the reviewer in the following file.

Reviewer 2 Report

Comments and Suggestions for Authors

Reviewer comments and suggestions

The study aimed to analyze the effect of 2 h inhalation of a krypton oxygen mixture (Kr 70%/O2 30%) on the recovery of neurological functions and the degree of brain damage in rats after photoinduced ischemic stroke (PIS). For this study, the authors used male Wistar rats and randomly assigned them into four groups: Two groups (n=20) underwent photoinduced ischemic stroke, followed by 2 hours inhalation of krypton-oxygen mixture Kr 70%/O2 30% or nitrogen-oxygen breathing mixture N2 70%/O2 30%, followed by neurological examinations on days 3 and 7, respectively; 2 groups (n=12) received only gas mixtures of the same concentration and exposure duration as in the PIS groups, The results of the study show that treatment with the krypton-oxygen mixture Kr 70 %/O2 30 % improves the neurological status on day 7 of observation, reduces the lesion volume according to the MRI examination. Furthermore, the study found that inhaling a krypton-oxygen mixture significantly increases the levels of phosphorylated AKT kinase (protein kinase B) and glycogen synthase kinase 3b (pGSK3b) and promotes the expression of transcription factor Nrf2, while suppressing the expression of transcription factor NFkB (p50).

Overall, the manuscript was well written. However, a few major concerns or comments needed to be explained or modified.

  1. In the introduction section please explain the reference 1 and 2 at their respective places, merely citation was not important
  2. The sentence is a little bit vague, please modify with the above following sentence “While xenon is widely used in clinical practice, researchers”
  3. what do the authors want to differentiate between brain injury and neuroprotection, check the third paragraph
  4. Comments for fourth paragraph “if this was the result, then the authors should write in the result part not in the introduction”.
  5. The line should be confusing while relating to the above line about GABA check the fifth paragraph of lines related to GABA
  6. Comments for Figure 1 Have the authors discuss the result of Figure 1 c
  7. The Figure 2 needs to be well arranged
  8. Comments for Figure 4 if the authors could provide the control or ladder consisting of western blot, that would be better
  9. Study limitations are also important to include in this study
  10. Please check the journal style of the references, all need to be modified based on the MDPI journal.

Author Response

Dear Editors, 

Thank you for giving us a chance to revise our manuscript. We have modified the text to answer all recommendations expressed in the reviewer’s comments. We have made the appropriate changes in the text and the figures. We appreciate the constructive criticism and believe that after these revisions, our manuscript will not raise more critique.

Thank you for your work.

We have attached the answers to the reviewer in the following file

Reviewer 3 Report

Comments and Suggestions for Authors

In the present manuscript, the authors demonstrated neuroprotective effects of krypton inhalation in vivo ischemic model rats. It was interesting but the experiments for mechanisms were insufficient.

In Abstract, conclusion was not presented. And, the authors wrote "promotes the expression of Nrf2, suppressing the expression of NF-kappaB (p50)" in the last sentence; however, the results did not show whether the increased Nrf2 expression decreased p50 expression, but only showed the increase of Nrf2 and decrease of p50.

In Introduction, "GABA inhibition" should be corrected. It confused to mean "inhibitory effects by GABA" or "inhibition of GABAergic neuron".

Materials and Methods, 2.3 Exposure to Krypton in Stroke+N2 or Kr groups, did "ninety minutes after PIS" mean infarct for 90 min and immediately exposure to gas mixture? In PIS model, could not reperfusion be performed?

Materials and Methods, 2.6 and 2.7, when were rat brains isolated after inhalation of gas mixtures? On day 14 same with MRI?

Results 3.2, figures shown in text were mistaken. They were not matched to the text content. In addition, P value in the text "the number of NeuN-positive cells in the penumbra" was different from the graph (Figure 3b; showing as number of neurons). Moreover, the graph for "the ratio of Iba 1-positive cells to lesion area" was lost in Figure 3.

Results 3.3, molecular mechanisms of neuroprotection were assessed when the animals were exposed to krypton for 2 hours. Was it immediately after krypton exposure? The data only showed the changes after krypton exposure compared to N2. Only the changes of molecules possible for neuroprotection might be selected by the authors. They might be associated with neuroprotective effects of krypton but the experimental evidence was not present. How about in Stroke+N2 and Stroke+Kr groups?

In Discussion, what was "PHI" model?

Author Response

(The authors gave the same response as above.)

Round 2

Reviewer 3 Report

Comments and Suggestions for Authors

The revised manuscript was improved according to the reviewer's comments.

The references [26] and [27] were not cited in text and the number [25] was out of order.

Author Response

Dear Reviewer, 

Thank you for giving us a chance to revise our manuscript. We have modified the text to answer all recommendations expressed in the reviewer’s comments.

Thank you for your work.

Below, we present the reviewer’s specific comments (in black) with our replies (in blue).

Sincerely,

Authors

The references [26] and [27] were not cited in text and the number [25] was out of order.

Answer: Please excuse this. We have checked all references and corrected the citations in the text.